

# Retrospective study on the correlation between CXCL13, immune infiltration, and tertiary lymphoid structures in cutaneous squamous cell carcinoma

Yulu Chen, Yuhao Wu, Zijun Zhao, Long Wen, Mingshun Wu, Dekun Song, Qingyu Zeng, Yeqiang Liu, Guorong Yan and Guolong Zhang

Department of Phototherapy, Shanghai Skin Disease Hospital, School of Medicine, Tongji University, Shanghai, China
Skin Cancer Center, Shanghai Skin Disease Hospital, School of Medicine, Tongji University, Shanghai, China

## ABSTRACT

**Background**. C-X-C motif chemokine ligand 13 (CXCL13) is a crucial chemokine for the recruitment of immune cells and the formation of tertiary lymphoid structure (TLS) in the tumor microenvironment. However, the relationship between CXCL13 and immune infiltration in cutaneous squamous cell carcinoma (cSCC) remains unclear.
**Objective**. We aimed to investigate the expression of CXCL13 and explore its association with immune activation and TLS in cSCC.
**Methods**. A total of 63 cSCC patients were involved in the present study. Hematoxylin and eosin staining was used for pathological examination of cSCC. Bioinformatics analyses and immunohistochemical staining were employed to access the expression of CXCL13 and TLS states. Public single cell RNA-sequencing atlas of skin disorders and multiplex immunofluorescence were used to explore CXCL13-producing cells.
**Results**. Utilizing the public database and our clinical cohort, we observed robust CXCL13 expression in cSCC tissues and a significant correlation with immune activation. Higher expression levels of CXCL13 were associated with lower histopathological grades and increased TLS formation. Furthermore, we confirmed that T cells and fibroblasts were the predominant cell types of CXCL13 secretion in cSCC.
**Conclusions**. CXCL13 is up-regulated in cSCC, which shows a significant positive correlation with immune infiltration and TLS formation. Our results underscore the role of CXCL13 in shaping the cSCC microenvironment, highlighting its potential as a therapeutic target.

## INTRODUCTION

Cutaneous squamous cell carcinoma (cSCC) is an epithelial malignancy originating from keratinocytes within the epidermis of the skin, characterized by invasiveness and early metastatic potential. It represents the leading cause of mortality among non-melanoma skin cancers, primarily attributable to prolonged exposure to ultraviolet B radiation

Corresponding authors
Guorong Yan,
guorongyan@tongji.edu.cn
Guolong Zhang,
glzhangtj@tongji.edu.cn

(280–315 nm) from sunlight (*Winge et al., 2023*). The metastatic rate of primary cSCC ranges from 1% to 4%, with disease-specific mortality primarily linked to metastasis, resulting in poor prognosis and 3-year overall survival (OS) rates ranging from 29% to 46% (*Hwang et al., 2024*; *Venables et al., 2019*).

While surgical resection remains the primary treatment, its efficacy against highly invasive cSCC is limited, often resulting in significant rates of recurrence and metastasis. The advent of tumor immunotherapy has ushered in significant changes, shifting focus towards tumor immunity. The tumor microenvironment (TME) consists of immune cells, fibroblasts, bone marrow-derived inflammatory cells, signaling molecules, and extracellular matrix components. It plays a crucial role in modulating tumor immune responses and influences tumor initiation, progression, and metastasis (*Binnewies et al., 2018*). Tertiary lymphoid structure (TLS) within TME plays a crucial role in enhancing anti-tumor immunity. These organized aggregates serve as hubs that facilitate the recruitment and localization of various immune cells, including T cells, B cells, and dendritic cells, thereby improving immune surveillance and activation against cancer (*Kinker et al., 2023*; *Sarti Kinker & DaSilva Medina, 2023*). Additionally, tumors with well-developed TLS are often associated with improved responses to immunotherapy, as these structures indicate a pre-existing immune response and a conducive environment for therapeutic intervention (*Helmink et al., 2020*). Thus, TLS is integral to enhancing immune responses and offers valuable insights for optimizing cancer treatment strategies.

C-X-C motif chemokine ligand 13 (CXCL13), also known as B-lymphocyte chemoattractant or B cell chemoattractant 1, recruits immune cells such as B cells, $CD4^+$ T cells, follicular T helper cells (Tfh), and dendritic cells (DC) *via* the CXCR5 receptor. It induces B cells to produce $LT\alpha1\beta2$, thereby reinforcing the formation of TLS through a positive feedback loop (*Deteix et al., 2010*; *Nayar et al., 2016*). CXCL13 promotes the organization of T and B cell compartments within TLS (*Wang et al., 2021*). Additionally, CXCL13 facilitates isotype switching, affinity maturation, and proliferation and differentiation of both B cells and T cells, thereby enhancing the immune response within TLS (*Hua et al., 2022*; *Wang et al., 2022*). CXCL13 expression has been correlated with TLS rates in esophageal squamous cell carcinoma, non-small-cell lung carcinoma, stomach adenocarcinoma, and malignant mesothelioma, and it has been suggested as a predictive marker for immune checkpoint inhibitor-responsive bladder cancer (*Groeneveld et al., 2021*). However, its relationship with TLS in cSCC has not been investigated.

To enhance our understanding of CXCL13 in cSCC, we analyzed both public databases and clinical data from Shanghai Skin Disease Hospital. This study aims to elucidate the relationship between CXCL13 expression and immune infiltration, assess its association with TLS, and explore its correlations with clinicopathological features in cSCC. Additionally, we seek to identify the primary cellular sources of CXCL13 within the TME. By integrating these analyses, we propose for the first time that CXCL13 may significantly influence TLS formation in cSCC and provide insights into its impact on immune responses and clinical outcomes.

## MATERIALS & METHODS

### TME deconvolution

A total of 120 RNA-sequencing data were downloaded from the Gene Expression Omnibus (GEO) with the accession code GSE199070. Then, 53 samples comprising 40 primary cSCC, one metastatic cSCC, and 12 normal sun-exposed skin (NS) were kept for cell composition deconvolution analysis. Abundance scores for 11 cell types (T cells, CD8[+] T cells, cytotoxic T cells, B cells, NK cells, monocytes, macrophages, myeloid dendritic cells, neutrophils, endothelial cells, cancer-associated fibroblasts) in the TME were quantified using MCP-counter (*Becht et al., 2016*). Since the lack of cSCC datasets with both transcriptomic and clinical information, survival analysis for CXCL13 was performed in head and neck squamous cell carcinomas using the GEPIA website (*Lu et al., 2024*).

### Patients and samples

This was a retrospective study, and written informed consent was obtained from patients. This study included 63 consecutive patients diagnosed with primary cSCC, confirmed through pathological examination, at Shanghai Skin Disease Hospital (Shanghai, China) between June 2017 and June 2019. All patients had not received chemotherapy, immunotherapy, or radiotherapy prior to surgery. The study was approved by the Ethics Committee of Shanghai Dermatology Hospital (2024-50). Comprehensive clinical and histopathological data were anonymized and systematically collected for each patient to ensure confidentiality.

### Pathological examination

Tumor tissues from all patients were surgically excised, fixed in formalin, and embedded in paraffin before being sectioned. Histological evaluation was conducted by two experienced pathologists following the World Health Organization (WHO) grading criteria. Hematoxylin and eosin (HE) staining was employed to assess tumor infiltration depth, subcutaneous extension, lymphatic and vascular metastasis, as well as peripheral nerve invasion.

### Immunohistochemical staining

All tissue samples were initially heated at 60 °C for 30 min to enhance tissue adherence to the sections. Subsequently, samples were deparaffinized in xylene, hydrated through a series of graded alcohol baths, and treated with 3% $H_2O_2$ for 15 min to quench endogenous peroxidase activity. Antigen retrieval was then achieved by incubation in EDTA (pH 9.0) at sub-boiling temperature, followed by gradual cooling to room temperature. Sections were further blocked with 5% BSA solution at room temperature for 30 min. Primary antibodies were applied and incubated overnight at 4 °C. The following day, sections were incubated with secondary antibodies for 15 min at 37 °C, followed by DAB staining for 3 to 8 min. Staining was halted by washing with $ddH_2O$. Finally, counterstaining with hematoxylin, dehydration through graded alcohols, and mounting of the sections were performed.

## Immunohistochemical evaluation

The CXCL13 score was quantified as the CXCL13-positive area ratio using ImageJ software. Based on the median CXCL13 score (0.932), patients were stratified into CXCL13 high-expression (CXCL13high) and low-expression (CXCL13low) groups. TLS was defined as aggregates of immune cells, predominantly composed of B cells surrounded by T cells. Immunohistochemical (IHC) staining for CD3 and CD20 was used to identify TLS-positive structures, characterized by $CD3^+$ T cells encircling $CD20^+$ B cells. TLS density was calculated as the medium number across 3 randomly selected high-power fields (HPFs) for each section. All assessments were independently performed by two pathologists. Significant discrepancies were resolved through review by two additional pathologists to reach consensus. Final results were further verified by an independent pathologist to ensure accuracy.

## Immunofluorescence

We used CD3 and CD20 staining to identify TLS. After deparaffinization in xylene and rehydration in alcohol, samples were treated with 3% H2O2, followed by antigen retrieval using EDTA (pH 9.0). Samples were blocked with 3% BSA, then incubated overnight at 4 °C with CD3 and CD20 primary antibodies (1:500; Servicebio, Hubei, China). The next day, after PBS washes, Alexa Fluor 488-labeled secondary antibodies (1:400; Servicebio, Hubei, China) were applied for 15 min at 37 °C. DAPI staining followed for 10 min in the dark, with autofluorescence quenching and sealing of coverslips in an anti-fade mounting medium.

We performed multiplex immunofluorescence (mIF) for CD4 (1:500, Abcam, Cambridge, UK), CD8 (1:100, Abcam, Cambridge, UK), $\alpha$-SMA (1:8000, Servicebio, Hubei, China), and CXCL13 (1:200, CST). After deparaffinization in xylene and rehydration in graded alcohol, antigen retrieval was done with EDTA (pH 9.0), followed by overnight incubation with CD4 primary antibodies at 4 °C. The next day, samples were treated with CD4 secondary antibodies (1:500, Servicebio, Hubei, China) for 50 min at room temperature, then with 647 reagent for 10 min in the dark at room temperature. The same procedure was repeated for $\alpha$-SMA (1:500, Servicebio, Hubei, China) and $CD8^+$ CXCL13 (CD8, 1:200; CXCL13, 1:300, Servicebio, Hubei, China), including respective primary and secondary antibody incubations. Finally, a DAPI staining solution and an autofluorescence quench agent were applied before mounting. Imaging was performed using an epifluorescence microscope (Nikon Eclipse C1, Nikon, Tokyo, Japan). CXCL13 expression was quantified as the CXCL13-positive area ratio, while CD4, CD8, and $\alpha$-SMA expression were quantified as the positive cell ratio.

## Analysis of CXCL13-producing cells

The single-cell RNA-sequencing data for skin from the Deeply Integrated human Single-Cell Omics data (DISCO, https://www.immunesinglecell.org) was used to investigate the CXCL13-producing cells. This skin atlas encompassed 18 different skin disorders, including cSCC, and more than 40 cell types were annotated.

## Statistical analysis

IBM SPSS statistics version 22.0 (IBM Corp., Armonk, NY, USA) was used for data analysis, and GraphPad Prism 9.5 was used for graphic plotting. Heatmap and correlation plots were visualized in R. Clinical characteristics of patients were expressed as counts and percentages. Pearson correlation analysis was used to evaluate the relationships between CXCL13 and TLS density, as well as between CXCL13 and CD4, CD8, and $\alpha$-SMA expression. Fisher's exact test was used to analyze the association between CXCL13 and clinicopathological characteristics of cSCC patients. $P$ value $<0.05$ was considered statistically significant.

## RESULTS

### CXCL13 is positively correlated with the immune infiltration

A total of 53 samples from the GEO database were kept for cell composition inference, which includes 40 primary cSCC, one metastatic cSCC, and 12 NS. Among the primary 40 cSCC, eight were well differentiated, 15 were moderately differentiated, 13 were poorly differentiated, and 17 cases were not categorized. Results showed that compared to the NS, cSCC exhibited significant enrichment of T cells, $CD8^+$ T cells, cytotoxic cells, B cells, NK cells, monocytes, and macrophages, while myeloid dendritic cells, neutrophils, endothelial cells, and cancer-associated fibroblasts showed decreased presence in primary cSCC (Fig. 1A). CXCL13 was notably enriched in primary cSCC and correlated with increased levels of T cells ($r = 0.4951$, $p = 0.0002$), $CD8^+$ T cells ($r = 0.4881$, $p = 0.0002$), cytotoxic cells ($r = 0.4212$, $p = 0.0017$), B cells ($r = 0.3050$, $p = 0.0264$), NK cells ($r = 0.5111$, $p = 0.0001$), monocytes ($r = 0.4194$, $p = 0.0018$), and macrophages ($r = 0.4194$, $p = 0.0018$) (Fig. 1B). Furthermore, survival analysis using in 517 cases of head and neck squamous cell carcinomas showed that patients with higher CXCL13 expressions were associated with better overall survival ($p = 0.0032$) (Fig. 1C). Collectively, this result indicates that CXCL13 is associated with immune infiltration in cSCC.

### CXCL13 expression and clinicopathological correlations in cSCC

We investigated the correlation between CXCL13 expression and clinicopathological characteristics in 63 patients diagnosed with cSCC. Results revealed that patients exhibiting higher CXCL13 expression levels tended to have better pathological grades, suggesting a potential association between elevated CXCL13 density and favorable prognosis in cSCC. Furthermore, CXCL13 density was found to correlate with patient age, with those in the high CXCL13 group being older than those in the low CXCL13 group. While age itself does not directly impact disease progression, these findings suggest a potential age-related role for CXCL13 in the pathogenesis of cSCC, meriting further investigation. However, our study did not find any statistically significant correlation between CXCL13 expression density and other clinicopathological features such as subcutaneous invasion, lymphovascular invasion, perineural invasion, tumor thickness, gender, and site of sun exposure (Table 1).

![PeerJ]

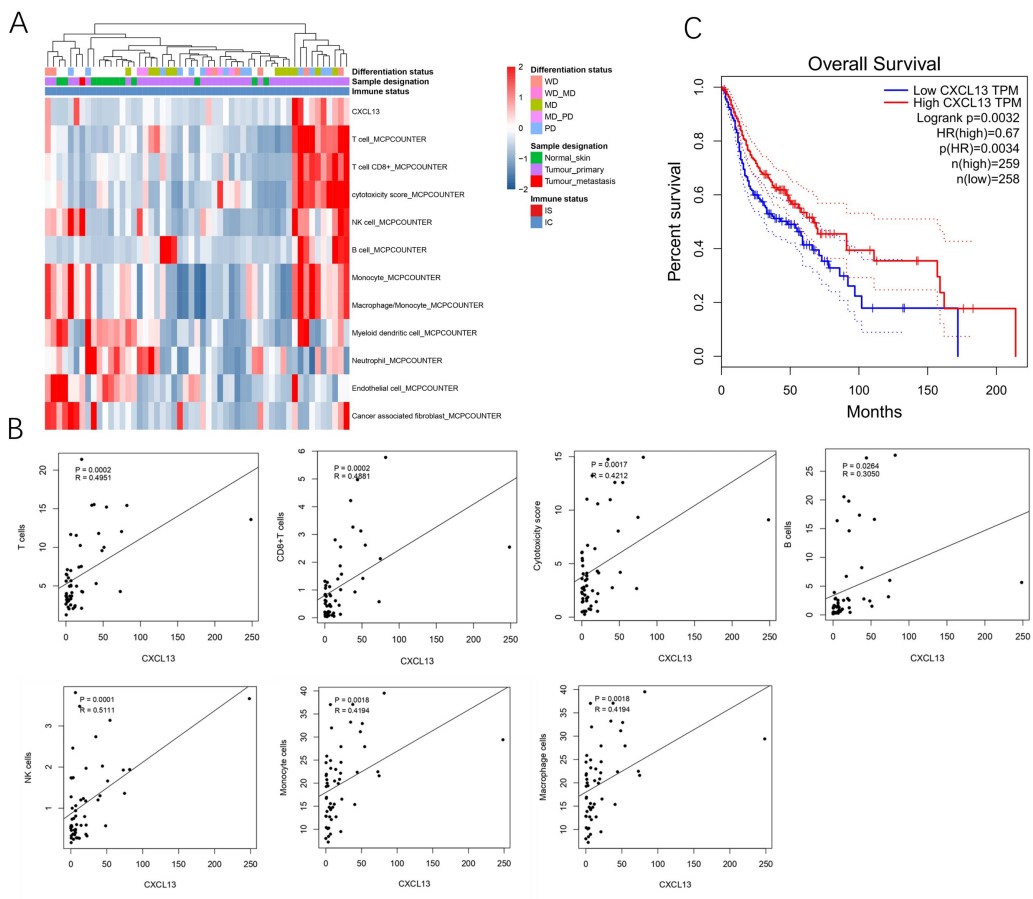

**Figure 1** **CXCL13 is associated with the immune infiltration.** (A) Tumor-infiltrating cells and CXCL13 distribution in normal skin and cSCC samples were assessed using the MCP-Counter method. (B) Correlation analyses of CXCL13 and T cells, CD8$^+$ T cells, cytotoxic score, B cells, NK cells, monocytes, and macrophages. (C) Survival analysis of CXCL13 in the head and neck squamous cell carcinoma.

## Association between CXCL13 expression and TLS was validated in clinical samples

CXCL13 is recognized as a marker for TLS (*Petitprez et al., 2020*). To explore the relationship between CXCL13 and TLS, we performed IHC analysis, which revealed CXCL13 positivity in 96.8% of all cSCC sections, indicating its ubiquitous presence in cSCC. CXCL13 expression was quantitatively assessed by calculating the positive area ratio at 100× magnification using ImageJ software. Based on the median percentage of CXCL13-positive cells (0.932), cases were stratified into CXCL13high and CXCL13low groups, comprising 47.6% and 52.4% of cases, respectively (Figs. 2A–2B).

TLS were defined as structures containing CD20$^+$ B cells surrounded by CD3$^+$ T cells at the periphery. TLS density was determined as the medium count across three randomly for each section (Fig. S1). TLS were identified in 69.8% of the samples (Figs. 2C–2E). Pearson correlation analysis was performed to evaluate the association between CXCL13 expression and TLS (Fig. 2F). The results demonstrated a significant positive correlation

**Table 1 CXCL13 Expression and Clinicopathological Correlations in cSCC.**

|  |  | Number | CXCL13$^{low}$ | CXCL13$^{high}$ | $\chi 2$ | *P*value |
|---|---|---|---|---|---|---|
| Histopathological grade | Poorly | 6(9.5) | 5(16.6) | 1(3.1) | 6.304 | 0.044[*] |
|  | Moderately | 12(19.1) | 8(26.7) | 4(12.1) |  |  |
|  | Well | 45(71.4) | 17(56.7) | 28(84.8) |  |  |
| Age | <65 | 13(20.6) | 10(33.3) | 3(9.1) | 6.296 | 0.039[*] |
|  | 65~89 | 42(66.7) | 18(60.0) | 24(72.7) |  |  |
|  | ≥90 | 8(12.7) | 2(6.7) | 6(18.2) |  |  |
| Subepithelial infiltration | negative | 61(96.8) | 29(96.7) | 32(97.0) | 0.005 | 1.000 |
|  | positive | 2(3.2) | 1(3.3) | 1(3.0) |  |  |
| Lymphovascular invasion | negative | 46(73.0) | 21(70.0) | 25(75.8) | 1.174 | 0.668 |
|  | positive | 17(27.0) | 9(30.0) | 8(24.2) |  |  |
| Perineural invasion | negative | 61(96.8) | 29(96.7) | 32(97.0) | 1.879 | 0.730 |
|  | positive | 2(3.2) | 1(3.3) | 1(3.0) |  |  |
| Thickness | <2 mm | 5(7.9) | 2(6.7) | 3(9.1) | 0.129 | 0.100 |
|  | 2–4 mm | 35(55.6) | 17(56.7) | 18(54.5) |  |  |
|  | >4 mm | 23(36.5) | 11(36.6) | 12(36.4) |  |  |
| Sex | Male | 33(52.4) | 13(43.3) | 20(60.6) | 1.880 | 0.132 |
|  | Female | 30(47.6) | 17(56.7) | 13(39.4) |  |  |
| Sun-exposed site | CSS | 47(74.6) | 23(76.7) | 24(72.7) | 1.049 | 0.701 |
|  | ISS | 9(14.3) | 3(10.0) | 6(18.2) |  |  |
|  | NSS | 7(11.1) | 4(13.3) | 3(9.1) |  |  |
| Maximum Diameter | <17 mm | 31(49.2) | 17(36.2) | 14(29.8) | 1.275 | 0.259 |
|  | ≥17 mm | 32(50.8) | 32(63.8) | 33 (70.2) |  |  |

**Notes.**

[*]$P < 0.05$.

CSS, continuously sun-exposed site; H-site, High-risk site; ISS, intermittently sunexposed site. Moderate-risk site; NSS, non-sun-exposed site.

The correlation between CXCL13 expression and clinicopathological features of CSCC was analyzed by chi-square test.

between CXCL13 expression levels and TLS presence ($r = 0.7126$, $p < 0.001$), providing strong theoretical support for further investigation into the role of CXCL13 in the tumor immune microenvironment.

## CXCL13 was mainly produced by fibroblasts and T cells

To elucidate the primary cellular sources of CXCL13 in skin diseases, we conducted an in-depth analysis of the ImmuneSingleCell skin atlas, a comprehensive database encompassing 18 distinct skin disorders, including cSCC, basal cell carcinoma, keloid, and psoriasis (Fig. 3A). This dataset profiles 40 cell types, such as epithelial cells, fibroblasts, CD8$^+$ T cells, CD4$^+$ T cells, Tregs, NK cells, and other immune cell populations (Fig. 3B). Subsequently, we investigated the expression patterns of CXCL13 within this dataset. Our analysis revealed that CXCL13 is predominantly expressed in CD8$^+$ T cells, CD4$^+$ T cells, and fibroblasts (Figs. 3C–3D).

To validate these findings, we performed mIF staining on clinical cSCC tissue samples. Fibroblasts were identified using alpha smooth muscle actin ($\alpha$SMA) as a marker, followed by mIF co-staining with CXCL13 (*Kalluri, 2016*). Triple staining for CD4, CD8, and

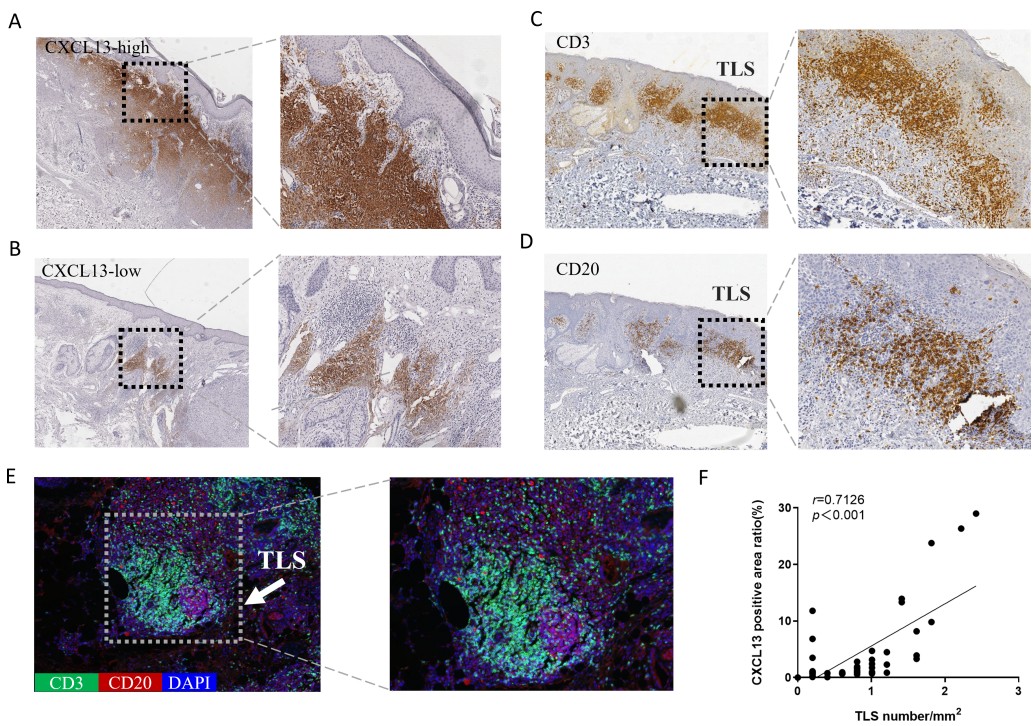

**Figure 2   Analysis of CXCL13 Expression and TLS in cSCC.** (A–B) Immunohistochemistry revealed CXCL13 positivity in 96.8% of all cSCC sections. Cases were categorized into CXCL13 high-expression (A) and low-expression (B) (40×, 100×). (C–E) Immunohistochemical and immunofluorescence analysis demonstrated the typical structure of TLS (composed of CD20 + B cells and CD3 + T cells), with TLS positivity observed in 69.8% of cSCC cases. (F) Pearson analysis demonstrated a significant positive correlation between CXCL13 expression and TLS density ($r = 0.713$, $p < 0.001$). CXCL13 scores were calculated as the median percentage of positive cells across five randomly selected 100× HPFs. TLS density was determined by counting the maximum number of TLS within three randomly selected HPFs at 40× magnification for each tissue section.

CXCL13 confirmed that CXCL13 is actively secreted by both CD8$^+$ T cells and CD4$^+$ T cells within these samples, consistent with our bioinformatic analysis (Figs. 3E–3F). For quantitative assessment, we selected four tissue sections each for CXCL13 co-staining with CD4$^+$ T cells, CD8$^+$ T cells, and cancer-associated fibroblasts (CAFs). In each section, five randomly selected HPFs at 100× magnification were analyzed to quantify the mIF staining results. Correlation analysis demonstrated strong and statistically significant associations between CXCL13 and CD4+ T cells ($r = 0.5046$, $p = 0.0233$), CD8$^+$ T cells ($r = 0.9318$, $p < 0.0001$), and CAFs ($r = 0.6500$, $p = 0.0019$) (Fig. 3G).

In conclusion, our integrated approach, combining public dataset analysis and clinical validation, demonstrates that CXCL13 is primarily produced by fibroblasts and T cells in skin diseases.

## DISCUSSION

In this study, we examined the role of CXCL13 in cSCC, with a particular focus on its relationship with immune infiltration and TLS. Our results demonstrated that CXCL13

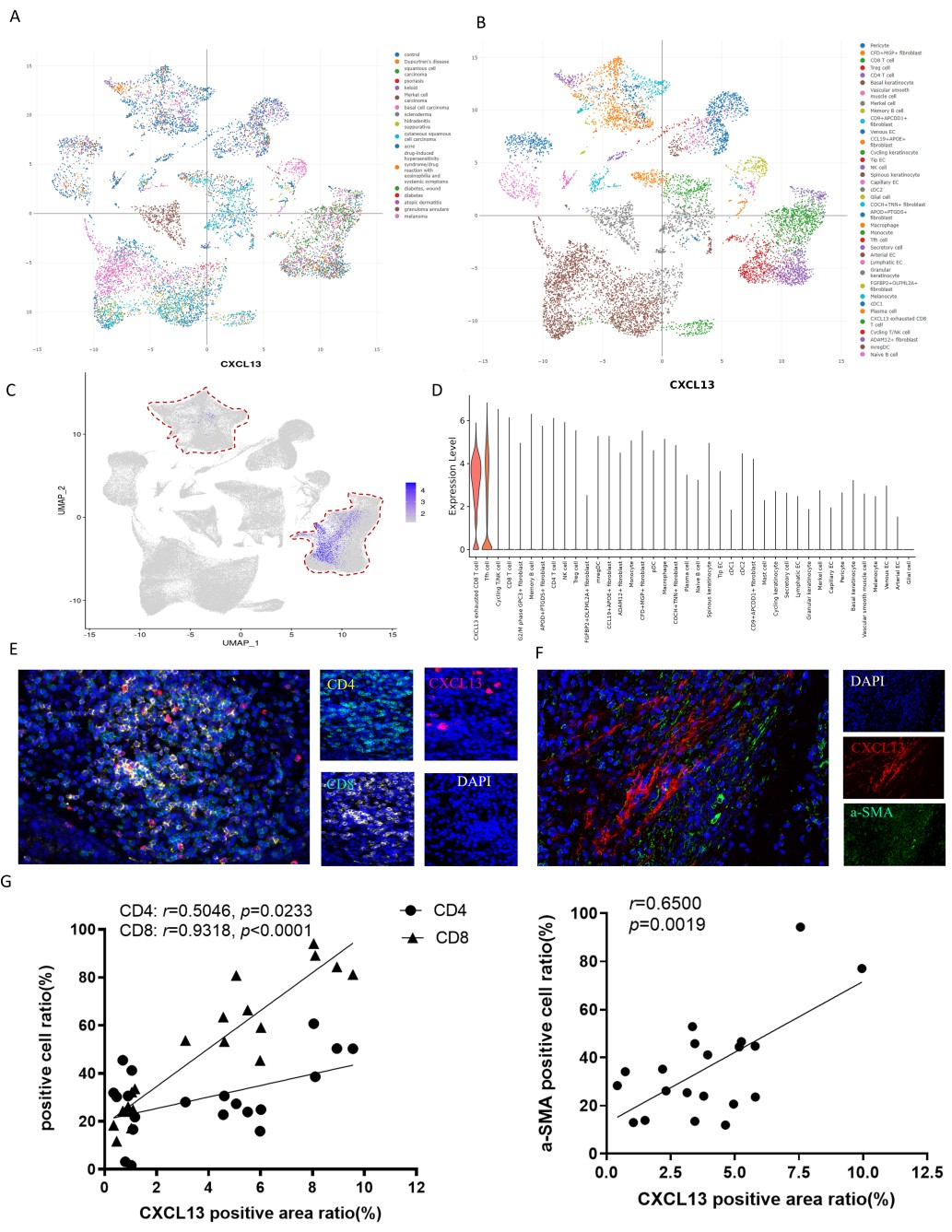

**Figure 3** **Identification of CXCL13 Cellular Sources in cSCC.** (A–B) The ImmuneSingleCell public database includes data from 18 skin diseases and 40 cell types in the skin atlas. (C–D) The ImmuneSingleCell public database identified predominant CXCL13 expression in exhausted CD8 + T cells and CD4 + T cells, with partial co-localization in CXCL13 + fibroblasts (indicated by red circles). (E-F) Spatial validation through multiplex immunofluorescence confirmed CXCL13 co-expression with CD4 + T cells, CD8 + T cells, and fibroblast markers in cSCC tumor microenvironments. (G) Correlation analysis showed strong and statistically significant associations between CXCL13 expression and CD4 + T cells ($r = 0.8270$, $p < 0.001$), CD8 + T cells ($r = 0.8035$, $p < 0.001$), and CAFs ($r = 0.7867$, $p < 0.001$).

is markedly overexpressed in cSCC tissues and shows a strong correlation with immune activation. Elevated CXCL13 levels are linked to an increased presence of TLS and lower histopathological grades, indicating a more organized and potentially less aggressive tumor microenvironment. IHC and mIF analyses identified T cells and fibroblasts as the primary sources of CXCL13 in cSCC. This underscores the critical role of CXCL13 in immune cell recruitment and TLS maintenance.

CXCL13, also known as B lymphocyte chemoattractant, is a key chemokine that directs immune cell migration and organizes immune responses (*Flippot et al., 2024*; *Ullah et al., 2024*). It primarily attracts CXCR5$^+$ cells, such as B cells and specific T cell subsets, to the sites of inflammation or tumorigenesis (*Reschke & Gajewski, 2022*). The binding of CXCL13 to its receptor CXCR5 activates multiple downstream signaling pathways, including the PI3K/AKT, MAPK/ERK, NF-$\kappa$B, and JNK pathways, as well as calcium signaling (*Liu et al., 2019*; *Wang et al., 2022*). These pathways collectively regulate critical cellular processes such as survival, proliferation, differentiation, inflammatory responses, and immune cell activation. Additionally, the local microenvironment and co-stimulatory molecules may further modulate the functional outcomes of CXCL13/CXCR5 signaling. By guiding these immune cells to the TME, CXCL13 plays a crucial role in the development and maintenance of TLS, thereby influencing the overall immune landscape and response within the tumor (*Rachidi et al., 2013*). By analyzing cSCC transcriptomic data from the GEO public database, we observed high CXCL13 expression in primary cSCC, which positively correlated with the infiltration of T cells, CD8$^+$ T cells, cytotoxicity score, B cells, NK cells, monocytes, and macrophages in primary cSCC. These results suggest that CXCL13 correlates with a more potent TME in cSCC. In high-grade serous ovarian cancer, elevated CXCL13 expression correlates with increased infiltration and activation of CXCR5$^+$ CD8$^+$ T cells within the TME. *In vitro* studies have shown that CXCL13 incubation promotes the proliferation and activation of CXCR5$^+$ CD8$^+$ T cells, which is associated with prolonged patient survival. The combined assessment of CXCL13, CXCR5, and CD8$^+$ T cells serve as an independent predictor of survival in high-grade serous ovarian cancer (*Yang et al., 2021*). In breast cancer, CXCL13 density positively correlates with disease-free survival and response to chemotherapy (*Gu-Trantien et al., 2017*). Similar findings are reported in gastric cancer, colorectal cancer, lymphoma, and non-small cell lung cancer (*Dai et al., 2021*). However, in clear cell renal cell carcinoma, elevated CXCL13 expression is associated with poorer progression-free survival and overall survival (*Xu et al., 2022*). In the present study, the correlation between CXCL13 and survival in cSCC was not observed may due to the limited sample size, we found elevated CXCL13 levels are associated with improved overall survival in head and neck squamous cell carcinoma. Furthermore, higher CXCL13 expression was linked to higher pathological grades and later onset ages. Therefore, we hypothesize that higher CXCL13 expression may suggest a more favorable prognosis.

CXCL13, produced by lymphoid tissue organizers, initiates TLS formation by recruiting lymphoid tissue inducer cells that further enhance CXCL13 secretion (*Li et al., 2023*). It promotes immune cell recruitment, compartmentalizes T and B cells within TLS, and upregulates LT$\alpha$1$\beta$2 expression in B cells, facilitating isotype switching and affinity maturation (*Hua et al., 2022*). Additionally, CXCL13 enhances the proliferation of

CXCR5$^+$CD8$^+$ T cells and increases IFN-$\gamma$ and granzyme B expression, strengthening the immune response in TLS (*Yang et al., 2021*). In our study, we observed high positive rates of CXCL13 (96.8%) and TLS (69.8%) in samples using IHC and mIF staining. A significant correlation between CXCL13 expression and TLS presence was identified, consistent with findings in other cancers such as esophageal squamous cell carcinoma, non-small-cell lung carcinoma, stomach adenocarcinoma, and malignant mesothelioma (*Brunet et al., 2022*). Studies have demonstrated that exogenous administration of CXCL13 can induce TLS formation in murine models (*Tang et al., 2016*). Collagen matrices containing LT$\alpha$1$\beta$2, CCL19, CCL21, CXCL12, CXCL13, and soluble RANK ligands have been shown to promote TLS formation and induce specific immune responses (*Kobayashi & Watanabe, 2016*). Research by *Delvecchio et al. (2021)* indicated that intra-tumoral injection of CXCL13 and CCL21 recruited CCR5$^+$ B cells and T cells into tumors, promoted TLS formation, and prolonged survival in pancreatic ductal adenocarcinoma mouse models. In preclinical models of ovarian cancer, intraperitoneal injection of recombinant CXCL13 could induce TLS formation, then inhibit tumor progression and enhance the efficacy of anti-PD-1 therapy through increased infiltration of CXCR5$^+$ CD8$^+$ T cells (*Delvecchio et al., 2021*; *Ukita et al., 2022*).

Utilizing data from the Immune Single Cell skin atlas, we identified CD4$^+$ T cells, CD8$^+$ T cells, and CAFs as the primary sources of CXCL13 in skin diseases. This was corroborated by our clinical samples through mIF, confirming these cell types as the origins of CXCL13. This observation aligns with findings in melanoma, non-small cell lung cancer, breast cancer, and hepatocellular carcinoma (*Li et al., 2019*; *Thommen et al., 2018*). Previous research has established TGF$\beta$1 and IL2 as pivotal inducers of CXCL13 expression in human blood CD4$^+$ T cells, facilitating the recruitment of CXCR5$^+$ T cells and B cells to the TME (*Workel et al., 2019*). Additionally, CAF secrete TNFR, a significant chemokine inducer of CXCL13 (*Rodriguez et al., 2021*). Nevertheless, further studies are essential to comprehensively elucidate these mechanisms.

## CONCLUSIONS

In conclusion, our study highlights CXCL13 as a key mediator of immune cell recruitment and TLS formation in cSCC, suggesting its critical role in promoting a more immunologically active TME. This may correlate with a less aggressive tumor phenotype. However, the relatively small sample size limits our ability to fully define the relationship between CXCL13 expression and patient prognosis, underscoring the need for further investigation with larger cohorts. Additionally, the absence of classical TLS in cSCC mouse models, due to species-specific immune differences, emphasizes the need for more refined *in vivo* models to better understand CXCL13-TLS dynamics in human contexts. Our findings deepen the understanding of CXCL13 in the immunobiology of cSCC and its potential prognostic significance. Clarifying CXCL13's role in TLS formation and TME modulation could pave the way for novel immunotherapies and precision oncology strategies, offering valuable prognostic and therapeutic insights not only for cSCC but also for broader oncological applications.

### Funding

This research was funded by the National Natural Science Foundation of China (82272761) and the National Key Research and Development Program of China (2023YFC2508200). The funders had no role in study design, data collection and analysis, decision to publish, or preparation of the manuscript.

### Grant Disclosures

The following grant information was disclosed by the authors:
National Natural Science Foundation of China: 82272761.
National Key Research and Development Program of China: 2023YFC2508200.

### Competing Interests

The authors declare there are no competing interests.

### Author Contributions

- Yulu Chen conceived and designed the experiments, performed the experiments, analyzed the data, prepared figures and/or tables, authored or reviewed drafts of the article, and approved the final draft.
- Yuhao Wu conceived and designed the experiments, performed the experiments, authored or reviewed drafts of the article, and approved the final draft.
- Zijun Zhao analyzed the data, prepared figures and/or tables, and approved the final draft.
- Long Wen analyzed the data, prepared figures and/or tables, and approved the final draft.
- Mingshun Wu analyzed the data, prepared figures and/or tables, and approved the final draft.
- Dekun Song analyzed the data, prepared figures and/or tables, and approved the final draft.
- Qingyu Zeng analyzed the data, authored or reviewed drafts of the article, and approved the final draft.
- Yeqiang Liu analyzed the data, prepared figures and/or tables, and approved the final draft.
- Guorong Yan conceived and designed the experiments, analyzed the data, prepared figures and/or tables, and approved the final draft.
- Guolong Zhang conceived and designed the experiments, authored or reviewed drafts of the article, and approved the final draft.

### Human Ethics

The following information was supplied relating to ethical approvals (i.e., approving body and any reference numbers):

The use of human samples for this study was approved by the ethics committee of Shanghai Skin Disease Hospital (2024-50).

## Data Availability

The raw measurements are available in the Supplemental Files. The sequences are available at GEO: GSE199070.

## Supplemental Information

Supplemental information for this article can be found online at http://dx.doi.org/10.7717/peerj.19398#supplemental-information.

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
