# Peer review of "Retrospective study on the correlation between CXCL13, immune infiltration, and tertiary lymphoid structures in cutaneous squamous cell carcinoma"

_PeerJ, doi:10.7717/peerj.19398_

## Round 0.1 · original submission · Major Revisions

Please address the comments of both reviewers and provide point by point responses.

·

Basic reporting

Yes the study is clear and unambiguous. The current condition is sufficient.

Experimental design

Yes, the current condition is sufficient.

Validity of the findings

Yes, the current condition is sufficient.

Additional comments

One of the most promising forms of treatment in cancer management is immunotherapy. This study is very interesting because it examines the role of CXCL13 which is the main mediator in initiating the immune process in carcinogenesis. Thus, it is necessary to understand and know the role of specific cytotoxic T cells associated tumors, so that they are able to carry out the process of eliminating tumor cells.


1. In line 124 of the manuscript, it is stated that immunohistochemical analysis was carried out using Image-J software. Please confirm whether Image-J or Image-J plus was used. If only Image-J was used, were there any other reasons stated by the author? Wouldn't it be better to use Image-J plus, which has higher sensitivity and specificity than regular Image-J?

2. In the study results mentioned in table 1 that most samples showed high degree in histopathology examination (71.4) however subepithelial infiltration (96.8) and perineural invasion (96.8) were negative results ? Would you like explain about that.

3. In the discussion it is mentioned that (line 238) : CXCL13, also known as B lymphocyte chemoattractant, is a key chemokine that directs immune cell migration and regulates immune responses. It primarily attracts CXCR5+ cells, such as B cells and certain subsets of T cells, to sites of inflammation or tumors. Please add to the discussion what specific downstream signaling pathways are activated by CXCR5 binding to CXCL13. Are there other factors that play a role?

Reviewer 2 ·

Basic reporting

In Figure 2, authors need to make it clearer what they are trying to convey in panels A-E. There are dotted squares, circles and arrows, but no indication of what they are highlighting in the images. Label D overlaps with one of the subfigures.

In figure 3A, point out within the figure the cell types that have CXCL13. The legend has many similar colors, and it is not easy to tell cell types from the legend alone. The IHC figures later in Figure 3 are not quantified at all.

Experimental design

In Figure 2F, instead of just high vs low, I want to see how the values of CXCL13 changes with TLS density in the form of a scatter plot. There need to be clear supplemental figures that show how TLS density is being assessed (where TLSs are encircled in the images).

Validity of the findings

There needs to much better quantification associated with the claim that CXCL13 is being secreted by CD4+ T cells, and CD8+ T cells. Furthermore, in the images in Figure 3, the red dots representing CXCL13 seem to also be present in cells other than just CD4 and CD8 cells. Please indicate through quantification that there are other cell types that seem to also be secreting CXCL13. Also, there is a label with CXCL12 in Figure 3, which does not get mentioned in the text anywhere?

Additional comments

While CXCL13 attracting cells like B and T cells is well accepted, the evidence presented in this paper is overall quite weak and needs considerable work. Images especially need to be quantified properly to support claims of better infiltration due to CXCL13.

---

## Round 0.2 · accepted · Accept

Authors have addressed all of the reviewers' comments and manuscript is ready for publication.

·

Basic reporting

Yes the current condition is sufficient.

Experimental design

Yes the current condition is sufficient.

Validity of the findings

Yes the current condition is sufficient.

Additional comments

The authors have responded to some of my concerns and explained and confirmed some of my questions about the manuscript. Revisions have been done adequately.

I have read the answers and corrections from the authors about the manuscript, I am satisfied with the explanation.

Thank you.

Reviewer 2 ·

Basic reporting

no comment

Experimental design

no comment

Validity of the findings

no comment

Additional comments

My previous reviews (Reviewer 2) requested additional clarity and quantification to justify the claims made in this paper. In their most recent update, it appears the authors have included requested quantification and also corrected the figures as I had requested. I believe the paper is now ready for acceptance. Good luck to the authors!